# *Atp11b* Deletion Affects the Gut Microbiota and Accelerates Brain Aging in Mice

**DOI:** 10.3390/brainsci12060709

**Published:** 2022-05-30

**Authors:** Cuiping Liu, Shibo Zhang, Hongwei Shi, Haicong Zhou, Junyi Zhuang, Yiyang Cao, Natalie Ward, Jiao Wang

**Affiliations:** 1Laboratory of Molecular Neural Biology, School of Life Sciences, Shanghai University, Shanghai 200444, China; ciciya@shu.edu.cn (C.L.); zhangshibo1995@shu.edu.cn (S.Z.); shihongwei1020@shu.edu.cn (H.S.); bluemaple@shu.edu.cn (H.Z.); junyi_zhuang@shu.edu.cn (J.Z.); 15239050331@shu.edu.cn (Y.C.); 2Banner Ocotillo Medical Center, 1405 S Alma School Rd, Chandler, AZ 85286, USA; natt.a.ward@gmail.com

**Keywords:** microbiota-gut-brain axis, brain aging, gut microbiota, *Atp11b*, aging pathology

## Abstract

The microbiota-gut-brain axis has attracted significant attention with respect to studying the mechanisms of brain aging; however, the specific connection between gut microbiota and aging remains unclear. The abnormal expression and mutation of proteins belonging to the P4-ATPase family, including *Atp11b*, results in a variety of neurological diseases. The results of our analysis demonstrate that there was a shift in the abundance of certain gut microbiota in *Atp11b*-knockout (KO) mice. Specifically, there was an increase in pro-inflammatory bacteria that accelerate aging and a decrease in probiotics that delay aging. Consequently, an enhanced oxidative stress response was observed, which was characterized by a reduction in the superoxide dismutase (SOD) activity and an increase in malondialdehyde (MDA) and reactive oxygen species (ROS) levels. In addition, our data demonstrate that there was a decrease in the number of cells in the dentate gyrus (DG) region of the hippocampus, and aggravation of aging-related pathological features such as senescence β-galactosidase (SA-β-Gal), p-HistoneH2AX (Ser139), and p16^INK4^. Moreover, KO mice show typical aging-associated behavior, such as memory impairment and slow pain perception. Taken together, we demonstrate a possible mechanism of aging induced by gut microbiota in *Atp11b*-KO mice, which provides a novel perspective for the treatment of aging through the microbiota-gut-brain axis.

## 1. Introduction

According to a report published by the United Nations, the global population over the age of 60 will reach 1.4 billion in the year 2030 and almost 2.1 billion in 2050, accounting for 21.3% of the world’s population [1]. Aging is a natural and time-related physiological phenomenon that is associated with a decline in overall bodily functions [2]. It is a slow and complex process involving multiple organs and systems [3]; however, the specific mechanisms underlying aging remain unclear.

The intestinal tract provides a conducive environment for microbial life and contains complex and diverse microbial communities. These microorganisms and their metabolites directly or indirectly affect many biological activities of the host, including nutritional processing, digestion and absorption, energy balance, immunity, and intestinal development and maturation [4]. In addition, changes in microbial composition affect the central nervous system by mediating the immune system, amino acid metabolism, the vagus nervous system, or intestinal nervous system. These imbalances lead to neurological diseases such as depression, Parkinson’s disease (PD), Alzheimer’s disease (AD), or autism spectrum disorder [5,6,7]; therefore, microbiota is a key regulator of gut-brain communication. Many studies have investigated the mechanism of the microbiota-gut-brain axis with a view to developing novel treatments for nervous system diseases [8]. It has been demonstrated that gut microbiota is associated with the aging process [9,10]. Aging has been linked to a decline in the diversity of gut microbiota, which is characterized by the accumulation of pro-inflammatory microorganisms and a decrease in beneficial microorganisms. This age-related gut microbiota status further accelerates aging [4]; thus, the intestinal tract is considered a key target organ for improving health in elderly animals and humans [11]. Previous data have shown that the transplantation of gut microbiota from young to old mice can decrease cognitive impairment, reverse differences in hippocampal metabolites, and reduce peripheral and brain inflammation, which can delay brain aging [12]. Moreover, it has been reported that related probiotics can prolong the lifespan of *Drosophila melanogaster* through microbiota-brain communication by preventing and treating neurodegeneration and cardiovascular disease, in addition to reducing physiological and oxidative stress [13]. Therefore, the microbiota-gut-brain axis plays a vital role in brain function, especially brain aging.

ATP11B is an important member of the P4-type phospholipid transferase (P4-ATPase) family. The ATP11B protein is responsible for phosphatidylserine (PS) and phosphatidyl ethanolamine (PE) translocation from the outer to the inner leaflet of the plasma membrane (PM), maintaining phospholipid asymmetry [14]. The normal functions of the ATP11B protein are vital in driving activities such as immunity and signal transduction [15]. Studies have shown that mutations in P4-ATPase can lead to neurodegenerative diseases, sensory deficiencies, and motor disorders [16,17]. In addition, our previous observations revealed that, in comparison with WT mice, *Atp11b*-KO mice are thinner and have a shorter life span. Defects in *Atp11b* disrupt the synaptic plasticity of the hippocampus in mice [18], which not only hinders the transduction of neural signals but also accelerates brain aging [19,20].

Given the important roles played by *Atp11b* in the aging process, it is reasonable to speculate that *Atp**11b* may participate in the interplay between the gut microbiota and brain aging. Here, we sequenced the 16S rDNA in the feces of WT and *Atp11b*-KO mice and analyzed the changes in gut microbiota using QIIME2 with ASV. Subsequently, we defined the functions of different bacteria and evaluated changes in the related pathological indexes in the brain. Our data highlight the function of *Atp11b* and reveal the mechanism underlying the interplay between changes in gut microbiota composition and aging.

## 2. Materials and Methods

### 2.1. Animals

The background of *Atp11b*-KO and WT mice was C57BL/6. All mice were reared at a constant temperature (22 ± 1 °C), humidity (60–80%), and 12 h/12 h light/dark cycle in a specified pathogen-free (SPF) facility at Shanghai University. Food and water were provided *ad libitum.* The mice were treated in accordance with the International Guidelines for Animal Research, and the study design was approved by the Animal Ethics Committee of Shanghai University (Approval No. ECSHU 2021-039, Date: 3 March 2021). Following extraction of genomic DNA from fecal samples, sequencing of the amplicons was performed on an Illumina MiSeq platform (Illumina, San Diego, CA, USA) at Majorbio Bio-Pharm Technology Co., Ltd. (Shanghai, China).

### 2.2. Cell Line

This study used the SH-SY5Y cell line, which was obtained from Shanghai Xuanya Biotechnology Co., Ltd. (Shanghai, China).

### 2.3. Amplicon Sequence Variants (ASV)

The high-quality sequences were denoised using the DADA2 plugin in the QIIME2 (version 2020.2) pipeline with the recommended parameters, which obtained single-nucleotide resolution based on the error profiles within samples. DADA2-denoised sequences are usually called amplicon sequence variants (ASVs). Taxonomic assignment of ASVs was performed using the naive Bayes consensus taxonomy classifier implemented in QIIME2 and the SILVA 16S rDNA database (v138). The confidence of the data is 0.7.

### 2.4. Rank-Abundance Curves

Rank-abundance curves were used to analyze diversity. To construct the curves, the number of sequences contained in each ASV in every sample were counted, the ASVs were sorted according to abundance (from large to small), and the sorting level of the ASV was used as the *x*-axis. The number of sequences contained in each ASV was used as the *y*-axis. The rank-abundance curve was used to explain two aspects of diversity: species richness and community uniformity. The richness of the species was demonstrated on the *x*-axis by the width of the curve; the larger the range of the curve, the higher the richness of the species. The shape (smoothness) of the curve was used to reflect the uniformity of the community in the sample; the smoother the curve, the more even the distribution of the species.

### 2.5. Principal Coordinate Analysis (PCoA)

Principal coordinate analysis (PCoA) is a non-binding data dimensionality reduction analysis method that is used to study the similarity or differences in sample community composition. A series of eigenvalues and eigenvectors were sorted and the most important eigenvalues were selected for use in the coordinate system. PC1 and PC2 were the two principal coordinate components: PC1 represented the component that explained the change in data as much as possible, whereas PC2 accounted for the largest proportion of the remaining degree of change. The analysis was performed using the R language software and employed the Bray-Curtis distance matrix algorithm.

### 2.6. Detection of Superoxide Dismutase (SOD)

The mice were anesthetized and sacrificed by cervical dislocation. The hippocampi were collected on ice and ground in SDD solution followed by centrifugation at 12,000 rpm, 4 °C for 3–5 min. The supernatant was collected for the assessment of SOD activity using a a Total Superoxide Dismutase Assay Kit (#S0101M, Beyotime, Shanghai, China) according to the manufacturer’s instructions.

### 2.7. Detection of Malondialdehyde (MDA)

Malondialdehyde (MDA) is a terminal product of lipid peroxidation during oxidative stress. The hippocampi were collected on ice and ground into a homogenate. The supernatant was collected by centrifugation at 12,000 rpm, 4 °C for 10 min. The supernatant was collected for assessment of the MDA level using an MDA detection kit (#S0131M, Beyotime, Shanghai, China) according to the manufacturer’s instructions.

### 2.8. Measurement of Intracellular Reactive Oxygen Species (ROS)

Intracellular ROS levels were measured using a ROS assay kit (#S0033S, Beyotime, Shanghai, China) based on 2′,7′-dichlorofluorescein-diacetate (DCFH-DA) according to the manufacturer’s instructions. DCFH-DA is easily oxidized to fluorescent dichlorofluorescein (DCF) by intracellular ROS, enabling quantitation of ROS levels by measuring fluorescence. Briefly, the cells were seeded on a confocal dish and transfected with the corresponding plasmid. After 36 h, the cells were incubated with DCFH-DA for 20 min at 37 °C and then analyzed by fluorescence microscopy (Carl ZEISS, Aalen, Germany).

### 2.9. H&E Staining

A hematoxylin staining kit (#C0105S, Beyotime, Shanghai, China) was used to stain the cells. Briefly, after deparaffinization and rehydration, the brain sections were stained with hematoxylin solution for 5 min followed by dipping 5 times in 1% acid ethanol (1% HCl in 70% ethanol). After rinsing in distilled water, the sections were stained with eosin solution for 3 min followed by dehydration in graded alcohol (95%, 90%, 80%, and 70%) and clearing in xylene. The coverslip was sealed and the cells were counted using a THUNDER Imager Tissue panoramic tissue imaging system (HistoFax SFII and HistoQuest Software, Sydney, Australia).

### 2.10. Senescence β-Galactosidase (SA-β-Gal) Staining

The brain slices were stained using a senescence β-galactosidase (SA-β-Gal) staining kit (#C0602, Beyotime) according to the manufacturer’s instructions. The film was sealed with anti-fluorescence quenching solution and analyzed under an inverted microscope (Nikon, Tokyo, Japan).

### 2.11. Immunofluorescence Assays

The brain sections were dried at room temperature and incubated with 10% FBS-PBS sealing solution at room temperature for 1 h. The sections were incubated overnight at 4 °C with the primary antibody, rabbit anti-p16^INK4^ (1:100, A5025, ABclonal, Wuhan, China), or rabbit anti-p-Histone H2AX (1:100, AP0687, ABclonal, Wuhan, China). The slides were washed 5 times with PBS (pH 7.4) for 3 min each and then incubated with goat anti-rabbit IgG H&L (ab150077, Abcam, UK) at 37 °C for 1 h. The slides were then washed 5 times with PBS (pH 7.4) for 3 min followed by incubation with DAPI solution for 5 min at room temperature. The slides were further washed 5 times with PBS (pH 7.4) for 3 min each and then mounted with anti-fluorescence quenching agent before covering with glass coverslips. The samples were analyzed by fluorescence microscopy (Carl ZEISS, Germany).

### 2.12. Morris Water Maze (MWM) Test

Prior to the MWM experiment, the mice were placed in the behavioral room for 3 days to adapt to the environment. The test was conducted in a circular tank (diameter 120 cm, height 80 cm). Water was injected into the tank at a depth of approximately 50 cm and the temperature was maintained at 22–25 °C. On day 1, the mice were placed into the water tank and allowed to freely acclimatize to the tank for 2 min. Each mouse was trained for 5 consecutive days, with four trials per day at 2 min per trial, and guided to climb onto the platform. On day 6, the platform was fixed, and the mice were allowed to move freely for 5 min. The behavior of each mouse was observed and recorded by the behavioral software (EthoVision XT 5.0, Beijing, China).

### 2.13. Novel Object Recognition (NOR) Test

The experimental device consisted of an opaque box and three objects (A1, A2, and B), where A1 was the same as A2 and B differed from A1 and A2. On day 1, the mice were trained to adapt to this device for 10 min. On days 3 and 4, the mice were placed in the device containing two objects (A1 and A2) and were allowed to freely explore for 5 min, after which, they were returned to the cage. An hour later, B replaced A1/A2 at the same location, and the mice were placed back into the device to freely explore for 5 min. The behavior of each mouse was observed and recorded by the behavioral software (EthoVision XT 5.0, Beijing, China).

### 2.14. Plantar Test

The infrared heat source was adjusted according to the plantar pain meter and placed under the plantar surface of the mouse. The time taken for the mouse to contract from the thermal stimulation was recorded as the contraction latency.

### 2.15. Pole Test

The pole test was performed in a quiet, dark environment. Mice were placed head-up on the top of a rough-surfaced vertical pole (diameter, 2.5 cm; height, 50 cm). The time taken to the first time-consuming head down and climbing up the pole was measured with a view to assessing the motor ability of the mice.

### 2.16. Statistical Analysis

Statistical analysis was performed using GraphPad Prism software (San Diego, CA, USA). Data are expressed as the mean ± SEM. Differences between two groups were assessed using an unpaired two-tailed Student’s *t*-test. In all statistical tests, the significance level was set at *p* < 0.05.

## 3. Results

### 3.1. Atp11b-KO Mice Display Suppressed Abundance of Gut Microbiota

To explore the changes in gut microbiota in *Atp11b*-KO mice, we analyzed the stool of two groups of mice by 16S rDNA microbial diversity sequencing. The sequencing results (Appendix A) were subsequently analyzed using QIIME2 [21]. The final results show that there were significant α and β diversity differences in the gut microbiota composition of the two groups of mice. Firstly, regarding the amplicon sequence variant (ASV) richness, WT mice had a higher gut microbiota abundance (*n* = 554) than *Atp11b*-KO mice (*n* = 244) (Figure 1A,B). In addition, 177 unique ASVs were identified in *Atp11b*-KO mice (24.21% of the total abundance) compared with 487 unique ASVs found in WT mice (66.62% of the total abundance) (Figure 1B). The Shannon indexes in the WT and KO groups were calculated at the ASV level. The results demonstrate that the gut microbiota abundance in KO mice was significantly lower compared with that in the WT group (Figure 1C), indicating a lower diversity of gut microbiota in KO mice. This suggests that *Atp11b* deficiency results in a significant reduction in gut microbiota communities. Principal coordinate analysis (PCoA) of β diversity based on the ASV level shows significant differences between WT and KO mice. Samples from each group were significantly clustered, indicating that the two groups were separated from each other (Figure 1D). These results demonstrate that *Atp11b* deletion altered the gut microbiota communities in mice, reducing the abundance and decreasing the diversity of microbial communities.

### 3.2. The Gut Microbiota Dysbiosis in Atp11b-KO Mice Is Associated with Aging

To further assess its role, we analyzed the differential gut microbiota in WT and *Atp11b*-KO mice based on the phylum, class, order, family, and genus level. At the phylum level, KO mice had a significantly higher proportion of Proteobacteria in comparison with WT mice. Proteobacteria drive the feed-forward circulation of intestinal degradation and inflammation, which, in turn, accelerates aging [22]. On the other hand, for Bacteroidota and Desulfobacterota, which promote digestion and health [23], there was a significant decrease in Desulfobacterota and a downward trend in the abundance of Bacteroidota (Figure 2A). At the class level, in comparison with WT mice, Gammaproteobacteria accounted for a larger proportion in KO mice, whereas Desulfovibrionia accounted for a smaller proportion (Figure 2B). A previous study has shown that Gammaproteobacteria contain many pathogens that cause enteritis and lung disease [24], and that Desulfovibrionia play an important role in nitrogen fixation and the maintenance of intestinal activity [25,26]. At the order level, there was a significant increase in the proportion of Pseudomonadales in KO mice, which could accelerate aging [27], whereas the proportion of probiotic Oscillospirales and Desulfovibrionales was significantly decreased (Figure 2C). At the family level, pathogenic bacteria Moraxellaceae and Aerococcaceae accounted for a large proportion in KO mice (Figure 2D). At the same time, the Circos map at the genus level demonstrates that the proportion of *Psychrobacter*, *Aerococcus*, and *Jeotgalicoccus* was higher in KO mice, whereas, in WT mice, the proportion of *Lactobacillus*, *norank_f__muribaculaceae*, *unclassified_f__Lachnospiraceae,* and *unclassified_f__prevotellaceae* was higher (Figure 2E). It should be noted that the three dominant bacteria in KO mice were pathogenic and that they mediate different diseases related to intestinal health and inflammation [28,29,30,31]. In contrast, the dominant bacteria in WT mice were probiotics, the lack of which often causes intestinal inflammation [32,33,34,35,36]. With an increase in age, gut microbiota in most mammals tends to result in a chronic low-degree pro-inflammatory state [37]. These results suggest that the altered structure of gut microbiota in *Atp11b*-KO mice may be related to aging.

### 3.3. Atp11b Deficiency Enhances the Oxidative Stress Response

Oxidative stress is a state of imbalance between oxidation and antioxidation, which is a negative effect of free radicals, and is a major factor leading to aging and disease [38]. Some studies have indicated that decreased Bacteroidota [39] and an abnormal increase in Proteobacteria [40] in gut microbiota may lead to excessive oxidative stress in mammals. Our sequencing data yielded similar results (Figure 2A). To ensure that *Atp11b* deletion not only leads to gut microbial dysbiosis but also causes aging-related characteristics, such as a high oxidative stress response, we analyzed the ROS levels, SOD activity, and MDA levels. The excessive accumulation of ROS causes oxidative stress reactions, which damage mitochondria and result in an energy crisis that triggers neurodegenerative diseases and accelerates aging [41]. We found a significant increase in the level of ROS in SH-SY5Y cells following *Atp11b* silencing (Figure 3A,B). SOD is able to scavenge and reduce excess free radicals in the body to delay aging [42]. The total activity of SOD in the hippocampus of KO mice was significantly decreased compared with WT mice (Figure 3C). In addition, the level of MDA in the hippocampus of *Atp11b*-KO mice was significantly increased (Figure 3D). MDA is a common lipid peroxidation product that accumulates during many pathophysiological processes and can be used as a reliable marker of oxidative stress [43]. These results suggest that *Atp11b* deletion weakens the antioxidant capacity and enhances the oxidative stress response; thus, we speculate that gut microbial dysbiosis caused by *Atp11b* deficiency may accelerate aging.

### 3.4. The Cell Number in the Hippocampal DG Region of Atp11b-KO Mice Is Reduced

Previous studies have demonstrated that the normal proliferation of cells and maintenance of the cell number in the brain are extremely important for normal brain functions [44]. A decrease in the number of cells accelerates the aging of the brain and is closely related to depression, cancer, AD, and other aging-related diseases [45]. To explore the effects of *Atp11b* deletion on the aging process of the brain in mice, brain slices from 3-month-old and 17-month-old WT and KO mice were stained with H&E (Figure 4A,C), and the number of cells in the hippocampal DG region was analyzed. The findings show that there was no significant difference in the number of DG cells between the 3-month-old KO and WT mice (Figure 4A,B); however, the number of cells in the 17-month-old KO mice was significantly decreased in the DG region (Figure 4C–E). These results demonstrate faster hippocampal aging in KO mice; therefore, we speculate that gut microbiota changes in KO mice may reduce the number of hippocampal cells, damage the hippocampal structure, and affect hippocampal functions, thus accelerating the aging process of the brain.

### 3.5. Aging-Related Pathology Is Aggravated in Atp11b-KO Mice

To further explore the effect of *Atp11b* on aging, aging-related pathology in the hippocampus of the mice was analyzed. SA-β-Gal is an important biomarker of aging and accumulates gradually during the aging of cells [46]. Our study shows that there was an increase in the SA-β-Gal level in the CA3 and DG regions of the hippocampus of KO mice (Figure 5A–C). Previous data have demonstrated that the phosphorylation of H2AX at Ser139 [p-H2A.X (ser139)] initiates dsDNA breakage and leads to aging-related inflammation and chronic diseases [47]. We analyzed the expression level of p-HistoneH2AX (Ser139) in the hippocampus of WT and KO mice (Figure 5D) and found it to be significantly upregulated in KO mice (Figure 5E). Moreover, p16^INK4^ is a reliable marker of aging [48]; therefore, we evaluated the expression level of p16^INK4^ in the hippocampus of WT and KO mice (Figure 5F) and found it to be significantly increased in KO mice (Figure 5G). These results further confirm our conjecture that the change in gut microbiota following *Atp11b* deletion leads to an increase in the expression of aging-related pathological markers in the mouse hippocampus, suggesting that *Atp11b* deletion accelerates the aging process of the brain by changing the dynamic balance of gut microbiota.

### 3.6. Aging-Associated Behaviors Are Displayed in Atp11b-KO Mice

Given the above gut microbiota sequencing and pathological results, we further investigated whether *Atp11b*-KO mice exhibit aging-associated behaviors. Typical symptoms of aging include learning and memory impairment and perception and movement retardation [49]. The MWM [50] and NOR [51] tests are recognized methods for the detection of learning and memory in mice. In addition, the plantar test can be used to evaluate sensitivity in mice, which reflects the perception and action ability of mice to a certain extent [52]. In the MWM experiment, the latency of the KO mice was significantly prolonged (Figure 6A), whereas the duration and frequency of exploration in the target quadrant was significantly reduced (Figure 6B,C). In the NOR test, the duration and frequency of novel object exploration in KO mice were significantly decreased compared with WT mice (Figure 6D,E), indicating that the deletion of *Atp11b* impaired learning and memory. Furthermore, KO mice displayed a longer contraction latency in the plantar test compared with WT mice (Figure 6F), indicating that KO mice were slower in perceiving the environment. To further evaluate the motor ability of the mice, the pole test was performed [53]. No significant difference was found between WT and KO mice with respect to the first time-consuming head down and climbing up the pole actions (Figure 6G,H), indicating that *Atp11b*-KO mice had no motor deficits. These results suggest that *Atp11b*-KO mice exhibited more serious aging-associated behaviors, and further implies that gut microbial dysbiosis caused by *Atp11b* deletion may accelerate aging via the gut-brain axis.

## 4. Discussion

In the present study, we found that *Atp11b* deletion decreased the abundance of gut microbiota in mice, indicating that *Atp11b* participates in the maintenance of gut microbial balance (Figure 7). Specifically, *Atp11b* deletion reduced the number of aging-related probiotics and increased the proportion of pathogenic bacteria (Figure 2). The deletion of *Atp11b* increased the aging-related oxidative stress response (Figure 3) and aggravated aging-related pathological changes, such as a decreased cell number (Figure 4) and increased expression levels of SA- β-Gal (Figure 5A–C), p-HistoneH2AX (Figure 5D,E), and p16^INK4^ (Figure 5F,G) in KO mice. In addition, severe aging-associated behavior was observed in KO mice (Figure 6). Taken together, these results suggest that the deletion of *Atp11b* may accelerate the aging process of mice by disturbing the abundance of gut microbiota.

Gut microbiota is a vital regulator of host brain health [54]. *Atp11b* deficiency results in gut microbial dysbiosis in mice, which is characterized by a reduction in microflora abundance and a shift in dominant bacterial species. We found that, in particular, changes in Proteobacteria (phylum level), Gammaproteobacteria (class level), and *Lactobacillus* (genus level) were dominant in *Atp11b*-KO mice. It has previously been reported that Proteobacteria can promote the inflammatory response and damage the nervous system [46], and a reduction in the abundance of Gammaproteobacteria and *Lactobacillus* can induce inflammatory reactions [55,56]. In addition, gut microbial dysbiosis leads to the secretion of proinflammatory molecules that disturb gastrointestinal and blood-brain barrier permeability [57], subsequently modulating the inflammatory signaling pathway that promotes neuroinflammation and neuronal injury and leads to neuronal death [57]. For example, Proteobacteria of the genus *Ralstonia* have been found at a significantly increased abundance in the mucosa of PD patients, which can trigger an inflammation-induced misfolding of α-synuclein [58]. Amyloid produced by different bacterial strains can aggravate AD pathology by promoting the misfolding of Aβ oligomers and fibrils [59]. Considering these data, we are led to assume that the deletion of *Atp11b* would alter dominant bacterial species in the gut, accelerating aging by aggravating brain inflammation.

Exosomes in the gut are involved in the regulation of immune cells, microbiota, and mucosal barriers [60]. Various P4-ATPases have been shown to contribute to intracellular vesicle formation and transport [61]. ATP11C has been suggested to alter membrane curvature by flipping phosphatidylserine onto the cytoplasmic leaflet of the trans-Golgi network (TGN), which provides a docking platform for proteins involved in vesicle production [62]. ATP11B interacts with vesicular transport proteins STX6 and VAMP4 to facilitate the transport of cisplatin from the TGN to secretory vesicles at the plasma membrane, suggesting its participation in exosome secretion [63]. Furthermore, ATP8B1 mediates the apical targeting of the TGN-derived sodium-dependent bile acid transporter SLC10A2/ASBT, which contributes to the reabsorption of bile salts in the gut [64]. These data suggest that the deletion of *Atp11b* may alter gut microbiota abundance by disturbing the exosome secretion or membrane localization of intestinal epithelial proteins.

In the present study, we also observed significantly increased levels of ROS and MDA and a decreased level of SOD in SH-SY5Y cells transfected with the sh*Atp11b* plasmid, indicating that the knockdown of *Atp11b* can induce increased oxidative stress. Oxidative stress is one of the pathogenic mechanisms for both neurodegenerative diseases and brain aging [65]. Gut microbiota regulates oxidative stress in the central nervous system via the microbiota-gut-brain axis [66]. It has been reported that probiotic *Lactobacillus paracasei* PS23 supplementation can delay age-related cognitive decline and elevate hippocampal SOD levels in mice [67,68]; therefore, these results indicate that *Atp11b* deletion can enhance oxidative stress, hence aggravating aging.

Microbial 16S rRNA gene sequencing allows for the simultaneous detection of dominant, rare, and unknown species in a sample, obtaining the microbial community composition as well as the relative abundance, and is the most commonly used method for the study of microbial composition [69]. Various statistical methods have been developed to perform differential abundance analysis based on abundance tables obtained from sequencing, which range from simple *t*-tests (Metastats) to zero-inflated Gaussian models (metagenomeSeq, RAIDA) and more complex log ratio statistics (ANCOM); however, these methods have some limitations, including the inability to adjust covariates and failure to use the complete information. Microbiome DDA, a generalized linear regression framework based on ZINB, has been developed with a view to increasing the robustness of count-based models [70]. In the future, we will use this algorithm to further explore the differences in species composition.

With an advancing biological age, the overall richness of gut microbiota tends to decrease, whereas bacterial taxa associated with unhealthy aging thrive, manifesting in gut microbial dysbiosis [71]. Thus, gut microbial dysbiosis can provoke innate pro-inflammatory immunity and other pathological changes that ultimately influence brain health [72]. Further investigation into the underlying mechanism of *Atp11b* deletion-induced gut microbiota changes will reveal more details regarding the microbiota-gut-brain axis.

## 5. Conclusions

This study shows for the first time that *Atp11b* deletion leads to gut microbial dysbiosis and aggravates aging-related pathology and behavior. Moreover, this study reveals the mechanism of aging and provides novel ideas for the prevention and treatment of aging-related diseases from the perspective of microorganism communities.

## Figures and Tables

**Figure 1 brainsci-12-00709-f001:**
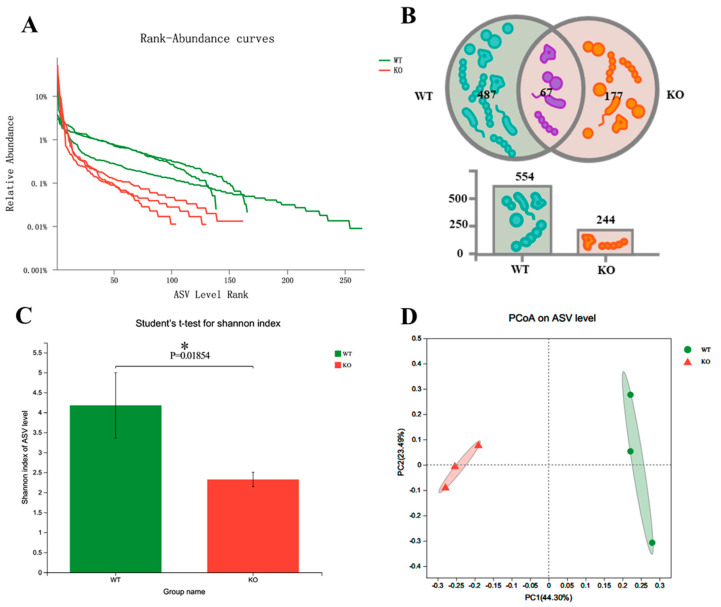
Diversity of gut microbiota in *Atp11b*-KO and WT mice. (**A**) Rank-abundance curves of WT (green) and KO (red) mice, representing the gut microbiota diversity. The *x*-axis indicates the ranking order of the number of ASVs, whereas the *y*-axis indicates the relative percentage abundance at the taxonomic level. (**B**) Venn diagram for the abundance of gut microbiota in WT and KO mice at the ASV level. (**C**) Community diversity of WT and KO mice according to the Shannon index. (**D**) PCoA of WT and KO mice based on the ASV level; a dot represents one mouse. For each group of mice, *n* = 3. Data are expressed as the mean ± SEM. Student’s *t*-test, * *p* < 0.05.

**Figure 2 brainsci-12-00709-f002:**
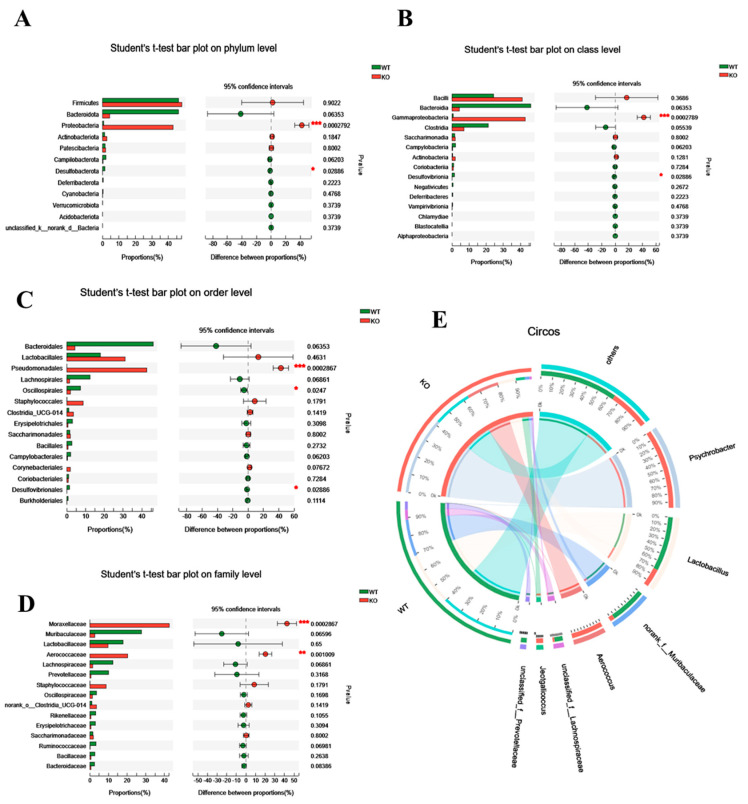
Differential abundance of gut microbiota between WT and KO mice. (**A**–**D**) Student’s *t*-test bar graph at the level of phylum (**A**), class (**B**), order (**C**), and family (**D**); the left side represents the gut microbiota category, and the right side represents the difference. (**E**) Circos genus relationship diagram. The left half-circle represents the genus composition in WT and KO mice, with the outer circle color representing WT (green) and KO (red). The right half-circle represents the proportion of distribution at the genus level. The color of the outer band represents different species, and the numbers on the inner circle represent the distribution proportion of a particular species in the WT (red) and KO (blue) groups. For each group of mice, *n* = 3. Data are expressed as the mean ± SEM. Student’s *t*-test, * *p* < 0.05, ** *p* < 0.01, *** *p* < 0.001.

**Figure 3 brainsci-12-00709-f003:**
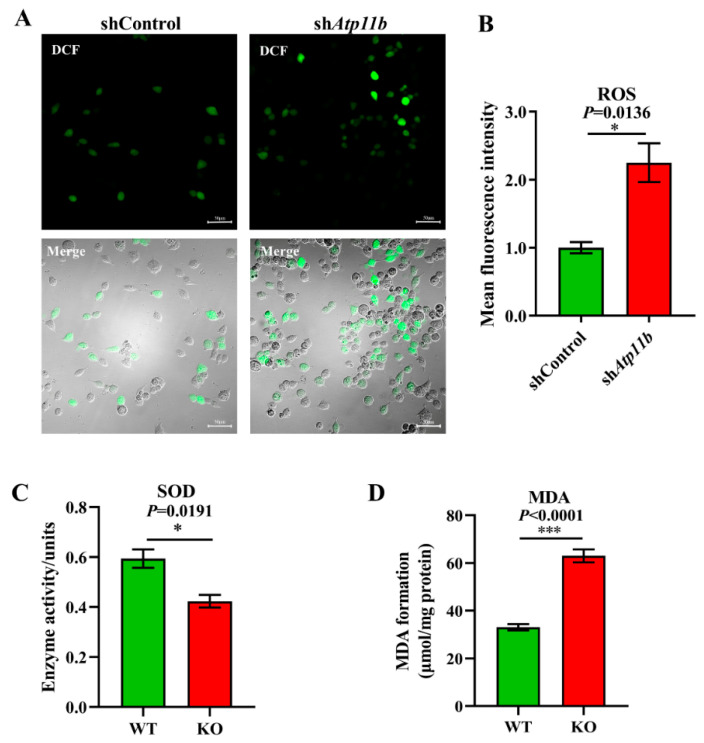
Deficiency in *Atp11b* weakens the antioxidant capacity and enhances the oxidative stress response. (**A**) SH-SY5Y cells were transfected with the corresponding plasmids, and, after 36 h, DCFH-DA was used as a probe to generate DCF for the quantitation of cellular ROS levels; scale bars: 50 μm. (**B**) The average fluorescence intensity of the generated DCF in SH-SY5Y cells of the two groups was calculated to measure the level of ROS. For each group of cells, *n* = 3. (**C**) Assay of the total SOD activity in the hippocampus of mice. For each group of mice, *n* = 3. (**D**) Determination of MDA levels in the hippocampus of mice. For each group of mice, *n* = 4. The mice were 12 months old. Data are expressed as the mean ± SEM. Student’s *t*-test, * *p* < 0.05 and *** *p* < 0.001.

**Figure 4 brainsci-12-00709-f004:**
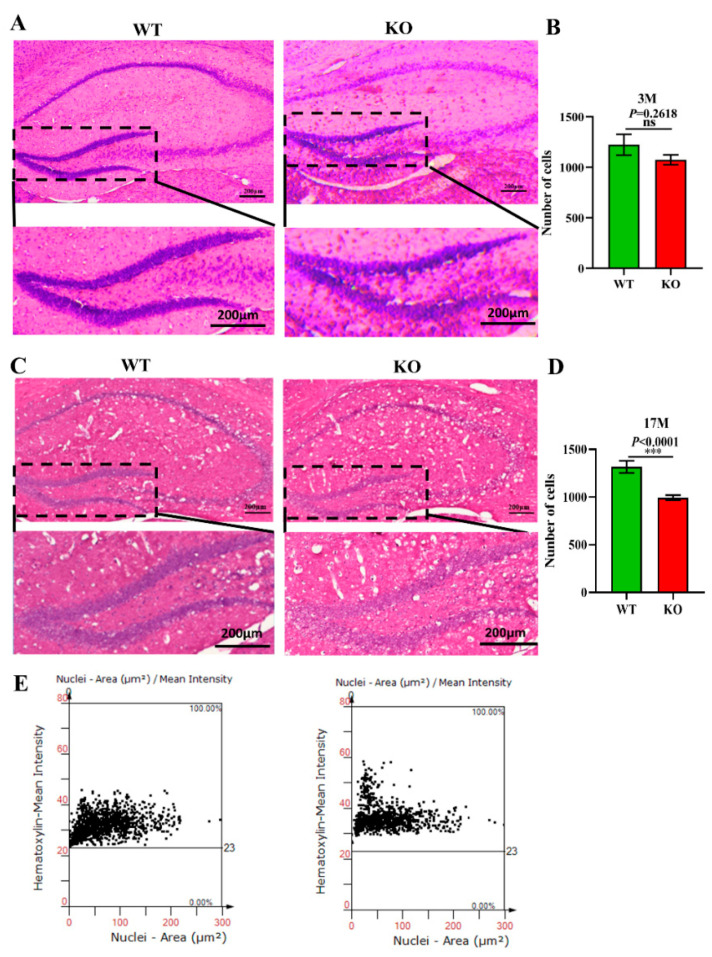
*Atp11b* deletion reduces the number of cells in the hippocampal DG region. (**A**) H&E staining of the hippocampus from 3-month-old WT and KO mice. (**B**) The number of cells in the DG region of 3-month-old mice. For each group of mice, *n* = 3. (**C**) H&E staining of the hippocampus in 17-month-old WT and KO mice. (**E**) The number of cells in the DG region of 17-month-old mice. (**D**) Scatter plot of the number of cells in the hippocampal DG region of 17-month-old mice. For each group of mice, *n* = 4; scale bars: 200 μm. Data are expressed as the mean ± SEM. Student’s *t*-test, *** *p* < 0.0001.

**Figure 5 brainsci-12-00709-f005:**
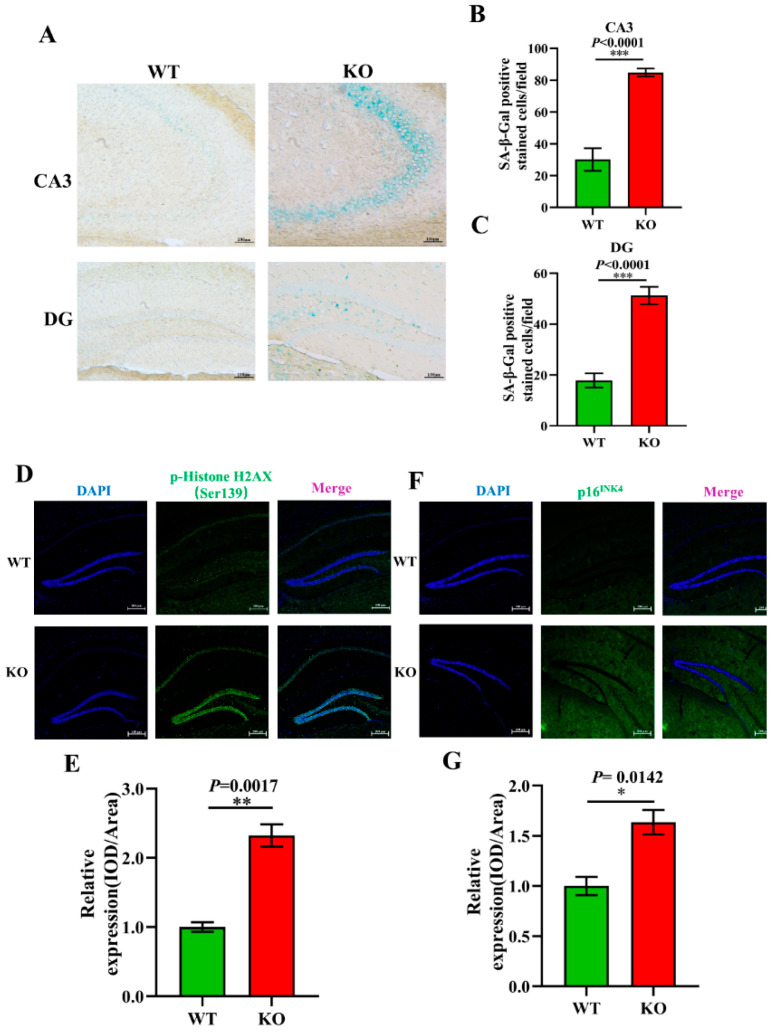
*Atp11b*-KO mice display aging pathology. (**A**) SA-β-Gal staining of the mouse hippocampus. (**B**) Quantitation of the SA-β-Gal-positive cells in the CA3 region. (**C**) Quantitation of SA-β-Gal expression in the DG region. Scale bars: 100 μm; for each group of mice, *n* = 4. (**D**) p-HistoneH2AX (green) fluorescence staining of the mouse hippocampus. (**E**) Average fluorescence intensity of p-HistoneH2AX in the mouse hippocampus. (**F**) p16^INK4^ (green) fluorescence staining in the mouse hippocampus. (**G**) Mean fluorescence intensity of p16^INK4^ in the mouse hippocampus. Scale bars: 200 μm; for each group of mice, *n* = 3. The mice were 12 months old; data are expressed as the mean ± SEM. Student’s *t*-test, * *p* < 0.05, ** *p* < 0.01, *** *p* < 0.001.

**Figure 6 brainsci-12-00709-f006:**
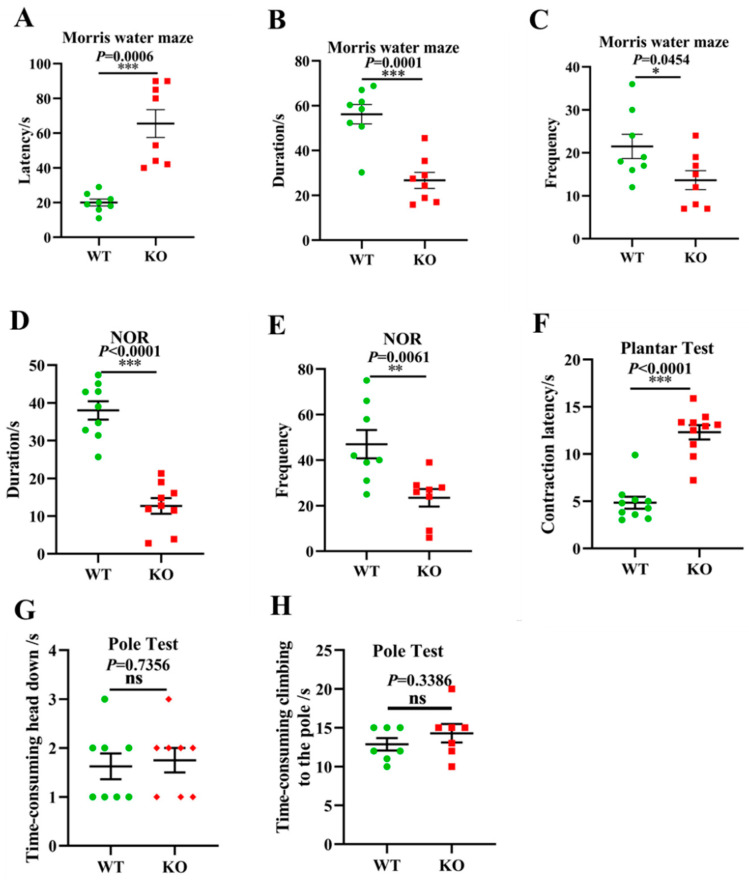
*Atp11b*-KO mice exhibit aging behavior. (**A**) The Morris water maze test showing the latency of mice. (**B**) The Morris water maze test showing the time that mice spent in the target quadrant. (**C**) The Morris water maze test showing the frequency that mice entered the target quadrant. (**D**) The NOR test demonstrating the time that mice spent exploring a novel object. (**E**) The NOR test demonstrating the frequency that mice explored a novel object. (**F**) Plantar test showing the contraction latency of mice. (**G**) The first time-consuming head down of mice in the pole test. (**H**) The time-consuming climbing up the pole of mice in the pole test. For each group of mice, *n* = 8–10. The mice were 12 months old. Data are expressed as the mean ± SEM. Student’s *t*-test, * *p* < 0.05, ** *p* < 0.01, *** *p* < 0.001.

**Figure 7 brainsci-12-00709-f007:**
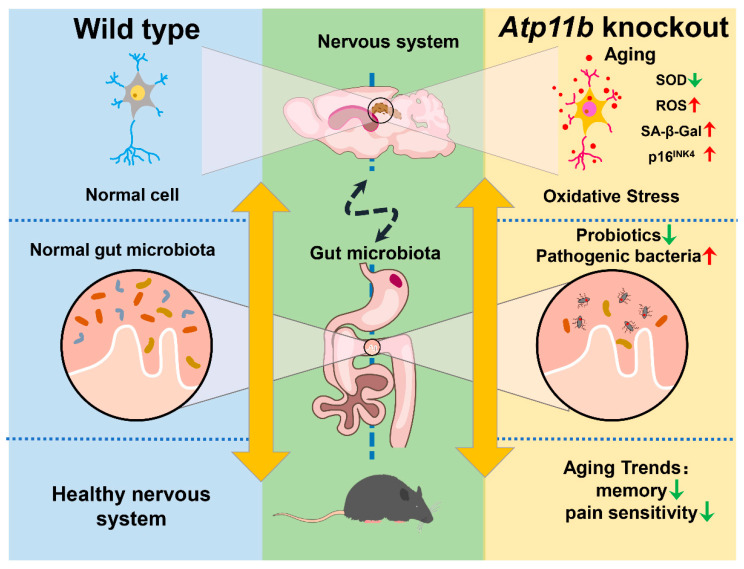
Schematic of the induction of gut microbial dysbiosis and acceleration of brain aging following *Atp11b* deletion. Gut microbial dysbiosis in *Atp11b*-KO mice accelerates brain aging, which manifests as an increased oxidative stress response in neurons and aging behaviors such as memory impairment and perceptual retardation.

## Data Availability

Not applicable.

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
