# Peer review of "Atp11b Deletion Affects the Gut Microbiota and Accelerates Brain Aging in Mice"

_brainsci, 2022, doi:10.3390/brainsci12060709_

Round 1

Reviewer 1 Report

In their manuscript, the authors suggest a role of the P4-ATPase flippase Atp11b in gut microbiota diversity and abundance, and aging-related pathology and behavior. They identify a distinct microbiota fingerprint in Atp11b knockout (KO) mice with predominance of bacterial pathogens associated with aging, and observe increased hippocampal expression of aging-associated markers in KO mice as well as aging-associated behavioral alterations, suggesting accelerated aging in the absence of Atp11b.

Overall, the manuscript is written clearly and comprehensively and the experimental approach is plausible. The authors provide novel insights into the microbiota composition, brain pathology and behavior in Atp11b KO mice. However, some important aspects need to be revised and clarified prior to publication.

Major points:

  1. To support their statement on enhanced oxidative stress response in Atp11b KO mice, the authors should confirm increased reactive oxygen species (ROS) levels in vivo by ROS immunostaining or flow cytometry in brain tissue of Atp11b KO mice. In the present version, ROS are only analyzed in vitro (Fig. 3).
  2. The authors propose reduced hippocampal cell numbers in Atp11b KO mice compared to controls. Details on the quantification process and applied software settings should be provided. It seems that the brain slices differ in the level of Bregma. To ensure comparability, the cell numbers should be normalized to the area of the region of interest. The maximal area of nuclei to be measured should be defined to count single cells only. Dots in Fig. 4B indicate cell sizes up to 400 µm2, which appears to high for individual cells. The images in Fig. 4B then need to be replaced accordingly and the legend of Figure 4B needs to be revised.
  3. The interpretation of the decreased hippocampal cell numbers as indicator for brain aging in KO mice will require assessment of hippocampal cell abundance in mice of at least two different age groups to exclude general developmental reduction in hippocampal cell numbers in the Atp11b mouse model or increased cell death.
  4. The discussion needs thorough revision. The authors need to discuss their results more in a broader context rather than listing their findings. Major questions are:
    1. How could the altered microbiota composition mechanistically drive aging pathology in Atp11b knockout mice?
    2. Are there other Atp11b-associated alterations such as reported blood-brain barrier leakage in Atp11b KO mice, neuroinflammation, immune activation and/or neurodegenerative processes that could cause brain aging?
    3. As Atp11b is linked to multiple functions, as mentioned in the introduction (lines 59-65), could there be direct effects of Atp11b on the brain and particularly neurons mediating accelerated aging independent of the microbiota? Of note, Atp11b is prominently expressed in neurons, astrocytes and endothelial cells (https://www.brainrnaseq.org/).
  5. Mutations in P4-ATPase genes were associated with motor deficits. Motor impairment of Atp11b KO mice should be evaluated to exclude effects on behavioral assessments due to decreased walking performance or increased freezing time.
  6. Throughout the manuscript, the authors draw causal conclusions between Atp11b deletion-associated gut microbiota composition and aging without providing data supporting causal evidence, such as fecal transplant experiments. Please revise sentences that overemphasize the causal role of the microbiota especially in lines 241, 246-247, 257-258, 275-276, 292, 331-332.

Minor points:

  1. The authors report microbiota differences between KO and wildtype mice and cite references that state sex-specificity in gut microbiota-mediated effects on aging. In the manuscript, details on the experimental animals are lacking. Please specify the age of the animals used in the different experiments and provide information on the animal background and housing of Atp11b KO and wildtype mice.
  2. It is difficult to read details in Figure 2 and I recommend providing graphs at higher resolution.
  3. Regarding section 3.3., the procedure of ROS quantification is not clear. Was the mean fluorescence intensity measured per area and was it normalized to cell numbers in the region of interest? Please provide data on transfection efficacy of the shAtp11b cells. Figure 3B: The scale bars needs to be adjusted for readability. Figure 3C: The y-axis should be labelled with “Mean fluorescence intensity” instead of “Mean gray value”.
  4. The scale bars in Figure 5A, D, F need to be increased for readability.
  5. In line 121, replace the word “There” by “The”.
  6. Remove sentences from the journals manuscript template in lines 176-178.
  7. In line 193, replace the word “distribution” by “diversity”. Distribution is misleading, as no data on the spatial distribution of microbiota in the gut are provided.
  8. In line 210, restructure the sentence to clarify that there is only a trend but no significance for decreased proportion of Bacteroidota and revise grammar for the last part of the sentence.

Author Response

Dear reviewer,

We are grateful for this chance to modify our manuscript. We thank you for the constructive criticism and valuable comments. We are sorry for the delay, which is due to Shanghai being under lockdown during the current severe situation of COVID19. Shanghai University has been locked down since March 2nd. To conduct the experiments suggested by the reviewers, we applied for several days in our laboratory, but the public experimental platform was not available. Due to the lack of express delivery and shutdown of biotech companies, we conducted the experiments using the stock reagents available in the lab.
Our manuscript was edited by Dr. Natalie Ward (Banner Ocotillo Medical Center, United States).
I sincerely appreciate all your time and effort.
Sincerely yours,
Jiao Wang, PhD.

Reviewer 2 Report

In this article, Liu et al. investigated the effect of Atp11B on gut microbiota in mice and brain aging. The topic itself is quite interesting. 

Major comments:

  1. Line 13: "decrease in the abundance of the gut microbiota", do authors mean the total biomass?
  2. Line85-88: which pipeline did the authors use here? QIIME1? Since the ASV approaches can provide a significant advantage for more precise identification of microbes, authors should use QIIME2 with ASV.
  3. Line 182-184: This is quite confusing.  What do you mean by OTU abundance is significantly different between KO and WT mice?
  4. Fig.1: Why the PCoA analysis was performed a the phylum level? Why not genus or OTU level? Also the exact P value should indicated in Fig1c/d.
  5. Line 203-Line230: The microbiome data is compositional data, differential abundance analysis should perform with some tools that can take those properties in. 
  6. Why only three mice in each group were selected? It seems that the sample size is too small given the personality of microbiome aon human and animal model.
  7. Line 271-284: It is still unclear how the gut microbiota changes affect aging process of the mouse brain? which functional pathway?

Author Response

Dear reviewer
We are grateful for this chance to modify our manuscript. We thank you for the constructive criticism and valuable comments. We are sorry for the delay, which is due to Shanghai being under lockdown during the current severe situation of COVID19. Shanghai University has been locked down since March 2nd. To conduct the experiments suggested by the reviewers, we applied for several days in our laboratory. 

Our manuscript was edited by Dr. Natalie Ward (Banner Ocotillo Medical Center, United States). And we hope the revised manuscript could be acceptable for you.

I sincerely appreciate all your time and effort.
Sincerely yours,
Jiao Wang, PhD.

Round 2

Reviewer 1 Report

The authors revised their manuscript according to the major and minor points suggested and provide a more comprehensive experimental approach by adding in vivo evidence for oxidative stress in Atp11b knockout (KO) mice, comparing hippocampal cell numbers in different age groups and analyzing motor ability Atp11b KO mice. They more profoundly discuss their results with respect to microbiota influence on accelerated brain aging in Atp11b KO mice. Particularly, including a schematic representation of the research findings in Figure 7 illustrates the overall statement.

Even though, there are some minor points that need to be added or revised prior to publication.

1.       Section 3.2: This section needs to be revised and significant/non-significant reports need to correspond to the re-analyzed data in Figure 2. Specifically, there is no specific decrease in Campilobacterota, but in Desulfobacterota on phylum level. On order level, there is no significant decrease in Campylobacterales, but in Oscillospirales.

2.       Figure 4B and D: Please show absolute cell numbers per area instead of % of WT cell number. Moreover, the images of the DG from 3 month-old and 17 month-old mice look as if they had undergone different processing and light exposure, which might also affect automatic cell counting. Please provide more similar images for all conditions and verify that counting procedure was not affected.

3.       Figure 6: Please include results from the pole test as provided in the letter to the reviewer to state comparable motor ability in WT and KO mice also in the manuscript.

Author Response

Dear Reviewer

We are grateful for this chance to modify our manuscript. We thank you for the constructive criticism and valuable comments, and we hope that the revised manuscript is now acceptable.

Sincerely yours,

Jiao Wang, PhD.

Reviewer 2 Report

1. Comment 1: This is totally wrong. The total biomass can only measure by some methods like qPCR or sequencing spike-in. The total sequencing count is not comparable across different samples at all. 

2. The title should be changed: for example, Atp11b deletion affects the gut microbiota and accelerates brain aging in mice.

3. Only genus, species or strain level names should be written in italic. 

4. Public all your related data, including 16S rRNA gene sequencing data.....

5. Comment4: the exact P- value should applied to all figures.

6. Comment5: the authors did not address my concerns at all.  Please do some literature searching for differential abundance analysis on microbiome data. There are branches of DDA methods, which can take compositionally and zero-inflated data. 

Author Response

Dear Reviewer,

We are grateful for this chance to modify our manuscript. We thank you for the constructive criticism and valuable comments, and we hope that the revised manuscript is now acceptable.

Sincerely yours,

Jiao Wang, PhD.
